# Novel Uracil-Based Inhibitors of Acetylcholinesterase with Potency for Treating Memory Impairment in an Animal Model of Alzheimer’s Disease

**DOI:** 10.3390/molecules27227855

**Published:** 2022-11-14

**Authors:** Vyacheslav E. Semenov, Irina V. Zueva, Sofya V. Lushchekina, Eduard G. Suleimanov, Liliya M. Gubaidullina, Marina M. Shulaeva, Oksana A. Lenina, Konstantin A. Petrov

**Affiliations:** 1Arbuzov Institute of Organic and Physical Chemistry, FRC Kazan Scientific Center of RAS, Arbuzov str. 8, Kazan 420088, Russia; 2Emanuel Institute of Biochemical Physics, Kosygina st. 4, Moscow 119334, Russia; 3Alexander Butlerov Institute of Chemistry, Institute of Fundamental Medicine and Biology, Kazan Federal University, Kremlyovskaya str. 18, Kazan 420008, Russia

**Keywords:** acetylcholinesterase, 6-methyluracil, inhibitors, peripheral anionic site, Alzheimer’s disease

## Abstract

Novel derivatives based on 6-methyluracil and condensed uracil, 2,4-quinazoline-2,4-dione, were synthesized with terminal *meta*- and *para*-benzoate moieties in polymethylene chains at the N atoms of the pyrimidine ring. In the synthesized compounds, the polymethylene chains were varied from having tris- to hexamethylene chains and quaternary ammonium groups; varying substituents (ester, salt, acid) at benzene ring were introduced into the chains and benzoate moieties. In vivo biological experiments demonstrated the potency of these compounds in decreasing the number of β-amyloid plaques and their suitability for the treatment of memory impairment in a transgenic model of Alzheimer’s disease.

## 1. Introduction

Alzheimer’s disease (AD), representing the most common cause of adult-onset dementia in people over 65 years of age, is characterized by progressive impairment in learning and memory at an early stage and ultimately results in death following progression of the disease [1]. The most studied pathological features of AD are β-amyloid (Aβ)-containing plaques and neurofibrillary tangles defined by the presence of microtubule-associated hyperphosphorylated tau protein [2].

Currently available therapies for AD are mainly aimed at palliating the symptoms of the disease by acting on the cholinergic system [3]. Acetylcholinesterase (AChE) and butyrylcholinesterase (BChE), which catalyze breakdown of the neurotransmitter acetylcholine in cholinergic synapses, are currently the most viable therapeutic targets for symptomatic treatment of AD. The rationale for the use of AChE and BChE inhibitors is the so-called “cholinergic hypothesis” [4,5]. Since the brains of AD patients are characterized by a loss of cholinergic neurons, partial inhibition of cholinesterase can compensate cholinergic deficiency to improve learning and memory [6,7]. However, in recent years, the AD drug-development pipeline has been dominated by research into disease-modifying therapies that can alleviate the progression of disease [8].

A growing body of evidence indicates that AD progression can be slowed by decreasing the number of soluble Aβ oligomers and Aβ plaques [9]. One of the promising avenues to achieve this is based on the following observation. Studies performed in recent decades indicate that the complex between AChE and Aβ has increased neurotoxicity [10] and that AChE itself additionally promotes oligomerization of Aβ fibrils through the interaction of Aβ with the AChE peripheral anionic site (PAS) [11].

Recently, a new class of selective mammalian dual binding site AChE inhibitors with an acyclic and macrocyclic structure based on 1,3-bis[ω-(substituted benzylethylamino)alkyl]-6-methyluracils has been reported [12,13,14]. Acyclic compounds, 1,3-bis[ω-(substituted benzylethylamino)alkyl]-6-methyluracils or quinazoline-2,4-diones, in which the substituted electron–withdrawing nitro-, trifluoro-, fluoro- and nitrile-group benzylethylamino moieties were bridged to N atoms of the 6-methyluracil or quinazoline-2,4-dione moiety via various polymethylene chains, demonstrated inhibitory power in the nanomolar range and selectivity for human AChE (hAChE) 10,000 or more times higher than for human butyrylcholinesterase (hBChE) [12,14]. The most active series of these compounds, **1**, along with nitro, trifluoro and nitrile substituents, is shown in Figure 1. The high activity and selectivity of the leading compounds was substantiated by molecular modeling and crystallographic data (PDB code 6F25) [13]; in particular, these compounds bound AChE as bifunctional inhibitors, blocking the entrance of the gorge of the enzyme with the substituted benzylethylamino group and masking the PAS area with the 1,3-bis(alkyl)-6-methyluracil moiety. In vivo experiments with APP/PS1 transgenic mice demonstrated improvement of the working memory and reduction of the number and area of Aβ plaques in the brain [12,13,14].

Compounds **1** contain electron–withdrawing nitro-, trifluoro, nitrilegroups in the *ortho* position at the benzene rings of the benzyl moieties [12,14]. While each of these groups has its own attributes in terms of the binding of compounds **1** with AChE, differences between compounds based on such substituents are not generally significant. Since compounds **1** are insoluble in water, the obvious way to make the substances soluble in water is by quaternization of atoms of N in the polymethylene chains. However, the quaternization of atoms of N in compounds **1** with nitrile substituents has been shown to result in a decrease in selectivity, along with a significant decrease in LD_50_ [14]. For this reason, it would be useful to be able to develop new methods for modifying compounds of a similar structure into water-soluble forms other than by quaternizing the N atoms with acid.

In this report, we investigate the extent to which activity towards AChE and BChE is affected by the introduction of a substituent, not in the *ortho* position at the benzene ring in compounds with the same structure as compounds **1** [12,14], but in other, *meta* or *para* positions. A related question concerns how such a change in the position of the substituent will affect the affinity for AChE and the selectivity for AChE vs. BChE. For this purpose, an electron-withdrawing ester group, specifically, the methoxycarbonyl group in the *meta* and *para* positions of the benzene ring, was chosen as a substituent. This ester group can be hydrolyzed to a carboxyl group in salt or acid form. Thus, in addition to the quaternization of the N atom in the polymethylene chains, the additional method can be used to modify compounds isostructural to compounds **1** into a water-soluble form.

Herein, a series of novel effective AChE inhibitors based on 1,3-bis[ω-(substituted benzylethylamino)alkyl]uracil derivatives is reported. These compounds, **2**–**4** (Figure 1), which are isostructural with compounds **1**, contain benzoate moieties with substituents in *meta* and *para* positions at the benzene ring within benzyl fragments. In the search of optimal structure for binding with the enzymes and elucidation of the structure–activity relationships, uracil derivatives were varied with the number of methylene units in the polymethylene chains. 6-Methyluracil and its condensed derivative, quinazoline-2,4-dione, were used as the uracil fragment. Chain lengths were varied from tris- to hexamethylene chains; these lengths seem to be optimal based on our previous studies. Thus, we present the synthesis of compounds **2–4**, which are evaluated in vitro and in vivo experiments in terms of binding with AChE and BChE, along with their potency to treat cognitive dysfunctions in animal models of AD.

## 2. Results and Discussion

### 2.1. Synthesis of 1,3-Bis[ω-(benzylethylamino)alkyl]uracils with Benzoate Moieties

The aimed-for compounds from series **2** and **3** (Figure 1) with 6-methyluracil and *meta*- and *para*-(3-bromomethyl)benzoate moieties were synthesized by starting from corresponding (3-bromomethyl)benzoate **7** and diamines **6a**–**d**, the latter of which were prepared from dibromides **5a**–**d** and ethylamine (Figure 1). This strategy was first developed for preparation of the compounds **1** and **2** [14], and can be used to vary uracil moiety, the lengths of polymethylene spacers, and the nature and position of substituents at the benzene rings of benzyl fragments within a wide range. In particular, by introducing into the synthetic procedure condensed uracil, quinazoline-2,4-dione derivatives, namely dibromides **8a**,**b** and diamines **9a**,**b**, the aimed-for quinazoline-2,4-diones **4a**,**b** were obtained (Figure 2).

Thus, 6-methyluracil and quinazolone-2,4-dione derivatives with *meta*-substituted methyl benzoate moieties (**2a**–**d**, **4a**,**b**) and 6-methyluracil derivatives with *para*-substituted methyl benzoate moieties (**3a**–**c**) were synthesized (Figure 2). Ester groups in the benzyl fragments, as expected, can be converted into a salt form. Therefore, as a result of boiling in MeOH with sodium alkali, the products of hydrolysis of esters **2c**, **3b** were isolated, namely disodium salts of benzoic acids **2e**,**d** with *meta*- and *para*-arrangement of carboxyl groups, respectively. Disodium salt **2e** passed through a column filled with the H-form of ion exchange resin qualitatively turned into diacid **2f**. Both salts and acid have moderate solubility in water. Additionally, the hydrobromination of compounds **2c** in MeOH produced water-soluble dihydrobromide **9** (Figure 2).

### 2.2. Inhibitory Activity towards Cholinesterases and Acute Toxicity of 6-Methyluracils and Quinazoline-2,4-diones

The inhibitory activities of compounds **2a**–**d**, **3a**–**d**, **4a**,**b**, and **9** against hAChE and hBChE were evaluated according to Ellman’s method [15]; the results for the IC_50_ values of all compounds are listed in Table 1 along with and their selectivity indexes for AChE over BChE. The acute toxicity of the compounds, discussed in terms of lethal doses (LD_50_) for mice, are also represented in the table.

For comparison with the compounds discussed, the table also demonstrates the data for inhibitors **1a**–**c** previously described [14]. The aimed compounds with *meta*-methyl benzoate moieties **2a**–**c** exhibited activity against AChE and BChE that clearly depends on the length of the polymethylene chains. Compound **2a** with trimethylene chains binds to AChE with an IC_50_ five orders of magnitude higher than the nanomolar concentrations, and its activity towards BChE is even higher than that towards AChE. The addition of methylene groups in the chains results in a sharp increase in the activity and selectivity against AChE. For compound **2b** with tetramethylene chains, the IC_50_ values are in the nanomolar range, and with the addition of one more methylene unit to the chains, compound **2c**, the IC_50_ values decrease by two orders of magnitude, while the selectivity towards AChE reaches a value almost equal to the selectivity of compound **1b** with *ortho*-trifluoromethyl substituents. With additional elongation of chains from 5 to 6 methylene groups, compound **2d**, no further increase in the activity and selectivity of the compound is observed; in particular, the inhibitory activity against AChE decreases slightly, while selectivity for AChE vs. BChE decreases by more than two orders of magnitude. The acute toxicity of *meta*-methyl benzoates **2a**–**d** was significantly lower than the toxicity of nitro- and trifluoromethyl compounds **1a**,**b** and the reference drug.

Quaternization of atoms of N in *m*-methyl benzoate **2b** with HBr acid, compound **9**, resulted in a decrease by several orders of both inhibitory power against AChE and in selectivity, along with a significant decrease in LD_50_. Conversely, another charged compound, namely disodium salt **2e**, demonstrated an increase by more than an order of magnitude in IC_50_ and more than 600-fold increase in selectivity compared to its neutral counterpart **2c**.

The targeted compounds with *p*-methyl benzoate moieties **3a**–**c** showed a slightly different affinity for AChE and BChE compared to *meta*-methyl benzoates **2a**–**d**. Maximum activity towards AChE and selectivity for AChE vs. BChE is achieved for compound **3c** with hexamethylene chains, and IC_50_ value is two orders of magnitude higher and the selectivity against AChE is more than 15 times lower compared to those for compound **2c**. As compared to ester **3b**, disodium salt **3d** showed the same changes in selectivity for AChE and acute toxicity as disodium salt **2e** compared to ester **2c**, in particular, a dramatic increase in selectivity towards the enzyme and increase in LD_50_. In contrast to counterparts **2c** and **2e**, the inhibitory power of salt **3d** against AChE increased by more than two orders of magnitude compared to counterpart **3b**. In general, these changes in the anticholinesterase activity of disodium salts in comparison with their counter-paired esters seem very interesting and worthy of further investigation.

Condensed uracils, quinazoline-2,4-diones with *meta*-methyl benzoate moieties **4a**,**b**, showed significantly less anticholinesterase activity than 6-methyluracil derivatives. When going from the 6-methyluracil to the quinazoline-2,4-dione moiety with the same chain length (compounds **2b** and **4a**, **2c** and **4b**), activity towards AChE sharply decreased by 2–3 orders of magnitude; moreover, there was less selectivity for AChE vs. BChE. The acute toxicities of compounds **2b** and **4a** are almost the same, while the toxicity of quinazoline-2,4-dione **4b** is higher than that of compound **2c**.

In general, the location of ester groups in *meta* positions at the benzene rings of the benzyl fragments seems preferable for anticholinesterase activity. 6-Methyl derivatives **2b-d**, **3c** were the most active against AChE in the series of the synthesized methyl benzoates. Taking into account the anticholinesterase activity of the compounds, namely affinity towards AChE, selectivity for AChE vs. BChE and low acute toxicity, *meta*-methyl benzoate **2c** is the leading compound in the series of synthesized inhibitors of AChE. Additionally, compound **2c** comprises a more effective and less toxic AChE inhibitor than previously synthesized inhibitors **1a-c** with *ortho*-nitro-, trifluoromethyl- and nitrile-substituents at the benzene rings of the benzyl fragments.

To study the mechanism of hAChE and hBChE inhibition by compound **2c**, the inhibition constants (Ki) were determined. Experiments were performed using three different concentrations of acetylthiocholine (ATC) as substrate for hAChE or butyrylthiocholine (BTC) for hBChE. Analysis of Dixon and Cornish-Bowden plots established that hAChE and hBChE inhibition was of mixed type, with Kci (competitive inhibition) and Kui (uncompetitive) components: Kci = 21−0 pM, Kui = 315 ± 95 pM for hAChE (Figure 3A,B), and Kci = 10.1 ± 3.1 µM, Kui = 24 ± 4.6 µM for hBChE (Figure 3C,D).

There is a possibility that the ester group of the compound **2c** can be hydrolyzed by AChE. To test this possibility, the compound **2c** (0.1 mM) was incubated overnight with AChE in phosphate buffer (pH 8.0). The same buffer containing compound **2c** but without AChE was used as a control. After the end of incubation, the buffer was filtered out of AChE using a centricon-30 ultrafiltration micro-concentrator from Amicon (Millipore Corporation, Billerica, MA, USA) and samples were analyzed by NMR spectroscopy. It was shown that the resonance signals both in the control spectrum and in the spectrum of the residue after AChE treatment are almost the same to those for compound **2c** (Appendix A). This means that both in the control and in the presence of the enzyme, the ester groups of the compound **2c** do not undergo notable hydrolysis to carboxyl groups, and there is no formation of hydrolysis products such as compounds 2e or **2f**. Thus, compound **2c** remains stable in the presence of AChE.

### 2.3. Molecular Docking Study of Lead Compound

The estimations of the p*K*_a_ values of the tertiary amine groups of compound **2c**, as obtained by Calculator Plugins of Marvin 21.14.0, ChemAxon [16], are 9.53 and 8.93, and 8.9 for both groups. Those obtained by MolGpka, a web-server using graph-convolutional neural network predictions [17,18], unanimously indicate that these groups are protonated at experimental pH. The position of compound **2c** obtained by molecular docking is similar to that obtained by X-ray crystallography for the analogous compounds **2c** [13]. Here appear π–cation interactions of one protonated amino group with side chains of aromatic residues of the gorge (Tyr337, Phe338 and Tyr341); π–π stacking interactions between the 6-methyluracyl ring and Trp286, supported by π–cation interactions with the other protonated amino group; T-stacking interactions of the benzyl ring with Trp86 indole ring. A key feature of binding mode of compound **2c** is the location of the COOCH_3_ group in the active site of AChE with the carbonyl oxygen atom in the oxyanion hole (Gly121, Gly122, Ala204) (Figure 4). Hydrolysis of esters by cholinesterases proceeds in two steps: acylation and deacylation [19,20]. At the acylation step a covalent bond between oxygen atom of the catalytic serine and carbonyl carbon atom of the substrate is form, this leads to formation of tetrahedral intermediate. The tetrahedral intermediate converts to acylenzyme due to proton transfer from the catalytic histidine to the ether oxygen atom of the substrate (Figure 5). However, the location of the ether oxygen atom in the docked position of compound **2c** makes proton transfer from His447 unlikely (Figure 5), along with hydrolysis of the methoxycarbonyl group *per se*. Favorable for hydrolysis orientation of the ligand is not possible in this case due to bulky methoxy part of the ester (Figure 5), compared to traditional substrates of AChE like acetyl- or butyryl- choline.

### 2.4. In Vivo Biological Assays

From the structure–activity profile of the compounds, compound **2c** seems to be most promising for inhibition of AChE in vivo. To test the ability of the compound **2c** to inhibit brain AChE in vivo, we measured AChE activity in brain homogenates following intraperitoneal injection of the compound **2c** at the LD_50_ dose. The brains were removed 30 min following injection of compound **2c**, and AChE activity in brain homogenates was measured spectrophotometrically according to the Ellman’s method [15]. It was shown that compound **2c** inhibits brain AChE activity on 52 ± 5% as compared to mean AChE activity in the brains of control mice. Thus, compound **2c** is of interest to study its suitability for the treatment of memory impairment in AD model.

During the first set of in vivo experiments, the ability of compound **2c** to reverse the scopolamine-induced amnesia in mice was studied by means of behavioral testing in a T-maze. The percentages of correct choices and mice reaching the criterion of learning were evaluated along with the dynamic of reaching the criterion of learning. Amnesia was caused by intraperitoneal administration of scopolamine (2 mg/kg) 40 min prior to testing. Donepezil (1 mg/kg) or **2c** (5 or 10 mg/kg) was daily orally administrated 20 min prior to test. The control group of mice was treated with saline. The behavior of mice was observed within 18 days (Figure 6A).

The scopolamine-injected mice had a diminished percentage of correct choices compared to the control mice (*p* < 0.001) (Figure 6B). The mice treated with compound **2c** at a dose of 10 mg/kg and donepezil at a dose of 1 mg/kg chose the correct direction in 75.47 ± 1.85% and 66.78 ± 2.62% of the tests, respectively. A lower dose of compound **2c** (5 mg/kg) had no therapeutic effect: mice chose the correct direction in 64.47 ± 1.47% (*p* < 0.05 compared to the control mice) (Figure 6B).

The scopolamine injection significantly reduced the reaching the criterion of learning at day 14: the percentage of animals achieving the criterion was 11% vs. 88% of the control group (*p* < 0.05) (Figure 6C). At both doses (5 and 10 mg/kg), donepezil and compound **2c** did not result in a significant decrease in the tasks learned by the animals. (Figure 6C). In the group of mice treated with 5 mg/kg of compound **2c**, 40% of mice reached the criterion of learning, whereas the percentage of task-learned mice treated with either 10 mg/kg of compound **2c** or 1 mg/kg of donepezil was 80% (Figure 6C).

Scopolamine-treated mice demonstrated a decline in the dynamic of reaching the criterion of learning. In contrast, mice treated with 10 mg/kg of compound **2c** or 1 mg/kg of donepezil showed a progressively increased trend of learning, comparable to the level of control group (Figure 6D).

Thus, memory deficit was improved by administration of either compound **2c** at a dose 10 mg/kg or donepezil at a dose 1 mg/kg. It is important to note that despite the higher affinity of compound **2c** against AChE compared to donepezil, the effective dose of donepezil in T-maze was significantly lower than the effective dose of the compound **2c**. This discrepancy between in vitro and in vivo efficacy may be explained by differences in pharmacodynamic or pharmacokinetic parameters of the compound **2c** and donepezil, such as different binding to blood proteins, different rates of metabolism or different ability to cross BBB.

During the second set of experiments, the effect of compound **2c** and donepezil on spatial memory performance in the AD transgenic mouse model (APP/PS1) was studied. Donepezil (1 mg/kg) and compound **2c** (10 mg/kg) were daily orally administrated 20 min before beginning of the test. Wild-type and APP/PS1 saline treated mice were employed as negative and positive controls, respectively. The behavior of mice in the T-maze was observed within 18 days (Figure 7A).

APP/PS1 mice demonstrated strong memory impairment in the T-maze, choosing the correct direction in 58.37 ± 2.35% of tests as compared to 69.12 ± 2.98% in WT mice (*p* < 0.05) (Figure 7B). Both compound **2c** and donepezil improved cognitive impairment in the APP/PS1 mice. The average percentage of choosing the right direction was 64.78 ± 2.83% in the group treated with compound **2c**; in the group treated with donepezil, the corresponding figure was 67.46 ± 2.35% (Figure 7B).

A significantly lower percentage of the APP/PS1 mice reached the learning criterion than that of the WT mice. The percentage of task-learned mice was 11% in APP/PS1 group and 75% in the group of WT mice (*p* < 0.05) (Figure 7C). Memory impairment in the APP/PS1 mice was significantly reduced by administrating the tested compounds: in the donepezil-treated group and the group treated with compound **2c**, the percentage reaching the criterion was 50% (Figure 7C).

In addition, as compared to WT mice, APP/PS1 mice demonstrated a delay in the dynamic of reaching the criterion of learning, which decreased by the administration of compound **2c** or donepezil (Figure 7D).

To test the possible disease-modifying action of compound **2c**, we studied the effect of compound **2c** and donepezil on the level of Aβ plaques in the brains of APP/PS1 mice using thioflavin S staining. Following the end of T-maze test, brains were cross-sectioned and stained with thioflavin S. The mean thioflavin S fluorescent spots per mm^2^ and the mean area of thioflavin S fluorescent spots indicating Aβ plaques in the mice entorhinal cortex were analysed. It was shown that administration of compound **2c** significantly decreased the Aβ plaque burden in the APP/PS1 mice (Figure 8A–C). Along with the mean Aβ plaque number, the mean Aβ plaque area was reduced by 36% as compared to the untreated control APP/PS1 mice (*p* < 0.001) (Figure 8B,C). However, important to note that donepezil did not significantly reduce the mean Aβ plaque number or mean area of plaques (Figure 8B,C). Thus, the disease-modifying effect of compound **2c** is significantly higher than the effect of donepezil.

## 3. Experimental Section

### 3.1. Chemistry

#### 3.1.1. General Methods

The NMR experiments were carried out on Bruker spectrometers AVANCE-400 (400.1 MHz (^1^Н), 100.6 MHz (^13^C)) and AVANCE-600 (600.1 MHz (^1^Н), 100.6 MHz (^13^C)). Chemical shifts (δ in ppm) were referenced to the solvents DMSO-d_6_ (δ = 2.50 ppm for ^1^H and 40.0 ppm for ^13^C NMR), CDCl_3_ (δ = 7.26 ppm for ^1^H and 77.0 ppm for ^13^C NMR. MALDI-TOF mass spectra were recorded on a Bruker ULTRAFLEX III mass spectrometer using *p*-nitroaniline as a matrix. Microanalyses of C, H, N were performed using a CHNS Vario Macro cube analyzer (Elementar Analysensysteme GmbH, Langenselbold, Germany; these were within ±0.3% of theoretical values for C, H, and N. Thin layer chromatography was performed on Silufol-254 plates (the solvent system was ethylacetate or ethylacetate/methanol mixture); visualization of spots was carried out under UV light (λ = 254 nm). For column chromatography, silica gel of 60 mesh from Fluka was used. All solvents were dried according to standard protocols.

Initial uracil and quinazoline-2,4-dione derivatives, particular dibromides **5a**–**d**, **8a**,**b**, bisamines **6a**–**d**, **9a**,**b** were prepared as previously reported [14].

#### 3.1.2. Synthesis of Cholinesterase Inhibitors, 1,3-Bis[ω-(benzylethylamino)alkyl]-6-methyl Uracils and Quinazoline-2,4-Diones with Methyl Benzoate Moieties

*Synthesis of compounds **2a**–**d**, **3a**–**c**, **4a**,**b***. *General procedure*: A mixture of bisamine **6a**–**d**, **9a**,**b** (3.0 mmol), methyl *meta*- or *para*-(bromomethyl)benzoate (6.0 mmol) and potassium carbonate (0.90 g, 6.5 mmol) was stirred in CH_3_CN (70 mL) at 60–65 °C for 12 h. After filtering out the precipitate, the solution was concentrated to 10–15 mL and transferred to a column with SiO_2_. The column was successively washed with petroleum ether, diethyl ether and 10:1 diethyl ether/methanol mixtures. The target compounds **2a**–**d**, **3a**–**c, 4a**,**b** were isolated from the 10:1 diethyl ether/methanol mixture fractions.

*1,3-Bis[3-(meta-methoxycarbonylbenzylethylamino)propyl]-6-methyluracil (**2a**)*: Yield 54%; oil; ^1^H NMR (CDCl_3_, 400 MHz) δ 7.97, 7.93 (both s, 1H each, 2H, Ar2H, Ar2′H), 7.90–7.86 (m, 2H, Ar5H, Ar5′H), 7.57–7.55, 7.50–7.48 (both d, 1H each, 2H, Ar4H, Ar4′H, *J* 7.6 Hz), 7.37–7.32 (m, 2H, Ar6H, Ar6′H), 5.54 (s, 1H, C5H), 3.93–3.90 (m, 2H, N3C7′H_2_),3.88 (s, 6H, 2COOCH_3_), 3.78–3.74 (m, 2H, N1C7H_2_), 3.58 (уш. s, 4H, C11CH_2_, C11′CH_2_), 2.53–2.46 (m, 8H, C9CH_2_, C9′CH_2_, 2NCH_2_CH_3_), 2.17 (s, 3H, C6CH_3_), 1.78–1.75 (m, 4H, C8CH_2_, C8′CH_2_), 1.05–1.02, 1.01–0.97 (both t, 3H each, 6H, 2NCH_2_CH_3_, *J* 7.2 Hz). ^13^C NMR (CDCl_3_, 100 MHz) δ: 167.7, 167.6 (COOCH_3_), 162.3 (C4), 152.4 (C6), 151.6 (C2), 141.0, 140.8, 134.0, 133.8, 132.7, 130.7, 130.5, 130.3, 130.2, 128.8, 128.7, 128.5 (Ar1-Ar6, Ar1′-Ar6′), 102.2 (C5), 58.3, 58.1 (C9, C9′), 52.6, 52.5 (C10, C10′), 51.1, 50.8 (NCH_2_CH_3_), 48.0, 47.2 (C7, C7′), 47.5 (COOCH_3_), 27.1, 25.6 (C8, C8′), 19.8 (C6CH_3_), 12.2, 12.1 (NCH_2_CH_3_). MALDI-MS (*m/z*): calcd for C_33_H_44_N_4_O_6_ [*M*-H]^+^ 591.3, found: 591.2. Anal. Calcd for C_33_H_44_N_4_O_6_: C, 66.87; H, 7.48; N, 9.45. Found: C, 66.80; H, 7.38; N, 9.51.

*1,3-Bis[4-(meta-methoxycarbonylbenzylethylamino)butyl]-6-methyluracil (**2b**)*: Yield 69%; oil; ^1^H NMR (CDCl_3_, 600 MHz) δ 7.94, 7.92 (both s, 1H each, 2H, Ar2H, Ar2′H), 7.88–7.85 (m, 2H, Ar5H, Ar5′H), 7.53, 7.50 (both d, 1H each, 2H, Ar4H, Ar4′H, *J* 7.2 Hz), 7.35–7.32 (m, 2H, Ar6H, Ar6′H), 5.51 (s, 1H, C5H), 3.88 (br. s, 8H, N3C7′H_2_, 2COOCH_3_), 3.74–3.71 (m, 2H, N1C7H_2_), 3.55 (уш. s, 4H, C11CH_2_, C11′CH_2_), 2.48–2.43 (m, 8H, C10CH_2_, C10′CH_2_, 2NCH_2_CH_3_), 2.16 (s, 3H, C6CH_3_), 1.60–1.58, 1.49–1.45 (both m, 4H each, 8H, C8CH_2_, C9CH_2_, C8′CH_2_, C9′CH_2_), 1.05–1.02, 1.01–0.97 (both t, 3H each, 6H, 2NCH_2_CH_3_, *J* 7.2 Hz). ^13^C NMR (CDCl_3_, 150 MHz) δ: 167.1, 167.0 (COOCH_3_), 162.0 (C4), 151.9 (C6), 150.9 (C2), 140.6, 140.4, 133.3, 133.2, 130.0, 129.8, 129.6, 128.1, 128.0, 127.9, 127.8 (Ar1-Ar6, Ar1′-Ar6′), 101.4 (C5), 57.7, 57.6 (C10, C10′), 52.8, 52.6 (C11, C11′), 51.9, 51.8 (NCH_2_CH_3_), 47.3, 47.1 (C7, C7′), 44.9 (COOCH_3_), 26.6, 25.4, 24.5, 24.4 (C9, C9′, C8, C8′), 19.5 (C6CH_3_), 11.6 (NCH_2_CH_3_). MALDI-MS (*m/z*): calcd for C_35_H_48_N_4_O_6_ [*M*]^+^ 620.4, found: 620.1. Anal. Calcd for C_33_H_44_N_4_O_6_: C, 67.72; H, 7.79; N, 9.03. Found: C, 67.80; H, 7.88; N, 8.97.

*1,3-Bis[5-(meta-methoxycarbonylbenzylethylamino)pentyl]-6-methyluracil (**2c**)*: Yield 72%; oil; ^1^H NMR (CDCl_3_, 400 MHz) δ 7.97, 7.95 (both s, 1H each, 2H, Ar2H, Ar2′H), 7.90–7.88 (both m, 2H each, 4H, Ar5H, Ar5′H, Ar4H, Ar4′H), 7.38–7.34 (m, 2H, Ar6H, Ar6′H), 5.53 (s, 1H, C5H), 3.90 (br. s, 6H, 2COOCH_3_), 3.88-(3.86 m, 2H, N3C7′H_2_), 3.76–3.72 (m, 2H, N1C7H_2_), 3.56 (br. s, 4H, C12CH_2_, C12′CH_2_), 2.52–2.46, 2.45–2.39 (both m, 4H each, 8H, C11CH_2_, C11′CH_2_, 2NCH_2_CH_3_), 2.18 (s, 3H, C6CH_3_), 1.63–1.55, 1.52–1.45, 1.36–1.27 (all m, 4H each, 12H, C8CH_2_, C9CH_2_, C10CH_2_, C8′CH_2_, C9′CH_2_, C10′CH_2_), 1.03–0.98 (m, 6H, 2NCH_2_CH_3_). ^13^C NMR (CDCl_3_, 100 MHz) δ: 167.1 (COOCH_3_), 162.1 (C4), 151.9 (C6), 150.9 (C2), 140.1, 133.5, 133.4, 130.1, 130.0, 129.9, 129.8, 128.3, 128.2, 128.1 (Ar1-Ar6, Ar1′-Ar6′), 101.3 (C5), 57.7, 57.5 (C11, C11′), 52.9, 52.8 (C12, C12′), 52.0, 51.9 (NCH_2_CH_3_), 47.3, 47.1 (C7, C7′), 45.0 (COOCH_3_), 28.7, 27.4, 26.5, 26.2, 24.7, 24.5 (C10, C10′, C9, C9′, C8, C8′), 19.6 (C6CH_3_), 11.5, 11.4 (NCH_2_CH_3_). MALDI-MS (*m/z*): calcd for C_37_H_52_N_4_O_6_ [*M*-H]^+^ 647.4, found: 647.5. Anal. Calcd for C_37_H_52_N_4_O_6_: C, 68.49; H, 8.08; N, 8.64. Found: C, 68.60; H, 8.00; N, 8.57.

*1,3-Bis[6-(meta-methoxycarbonylbenzylethylamino)hexyl]-6-methyluracil (**2d**)*: Yield 64%; oil; ^1^H NMR (CDCl_3_, 400 MHz) δ 7.97, 7.95 (both s, 1H each, 2H, Ar2H, Ar2′H), 7.90–7.88, 7.55–7.52, 7.38–7.35 (all m, 2H each, Ar5H, Ar5′H, Ar4H, Ar4′H, Ar6H, Ar6′H), 5.56 (s, 1H, C5H), 3.90 (br. s, 6H, 2COOCH_3_), 3.87–3.84 (m, 2H, N3C7′H_2_), 3.76–3.72 (m, 2H, N1C7H_2_), 3.57 (уш. s, 4H, C13CH_2_, C13′CH_2_), 2.49–2.42, 2.40–2.37 (both m, 4H each, 8H, C12CH_2_, C12′CH_2_, 2NCH_2_CH_3_), 2.20 (s, 3H, C6CH_3_), 1.63–1.54, 1.49 (both m, 4H each, 8H, C8CH_2_, C8′CH_2_, C11CH_2_, C11′CH_2_), 1.41 (m, 8H, C9CH_2_, C10CH_2_, C9′CH_2_, C10′CH_2_), 1.04–0.99 (m, 6H, 2NCH_2_CH_3_). ^13^C NMR (CDCl_3_, 100 MHz) δ: 167.2, 167.1 (COOCH_3_), 162.2 (C4), 151.9 (C6), 150.9 (C2), 140.7, 133.4, 133.3, 130.0, 129.9, 129.8, 129.7, 128.1, 128.0 (Ar1-Ar6, Ar1′-Ar6′), 101.3 (C5), 57.7, 57.6 (C12, C12′), 53.2, 53.0 (C13, C13′), 52.0, 51.9 (NCH_2_CH_3_), 47.3, 47.2 (C7, C7′), 45.1 (COOCH_3_), 28.9, 27.5, 27.1, 26.9, 26.8, 26.6, 25.6, 25.5 (C11, C11′, C10, C10′, C9, C9′, C8, C8′), 19.6 (C6CH_3_), 11.7, 11.6 (NCH_2_CH_3_). MALDI-MS (*m/z*): calcd for C_39_H_56_N_4_O_6_ [*M*]^+^ 676.4, found: 676.1. Anal. Calcd for C_39_H_56_N_4_O_6_: C, 69.20; H, 8.34; N, 8.28. Found: C, 69.11; H, 8.42; N, 8.17.

*1,3-Bis[4-(para-methoxycarbonylbenzylethylamino)butyl]-6-methyluracil (**3a**)*: Yield 72%; oil; ^1^H NMR (CDCl_3_, 400 MHz) δ 7.96–7.92, 7.39–7.36 (both m, 4H each, 8H, Ar3H, Ar5H, Ar3′H, Ar5′H, Ar2H, Ar6H, Ar2′H, Ar6′H), 5.52 (s, 1H, C5H), 3.90–3.87 (m, 8H, N3C7′H_2_, 2COOCH_3_), 3.75–3.72 (m, 2H, N1C7H_2_), 3.56 (уш. c, 4H, C11CH_2_, C11′CH_2_), 2.51–2.42 (m, 8H, C10CH_2_, C10′CH_2_, 2NCH_2_CH_3_), 2.16 (s, 3H, C6CH_3_), 1.65–1.57, 1.52–1.44 (both m, 4H each, 8H, C8CH_2_, C9CH_2_, C8′CH_2_, C9′CH_2_), 1.02–0.99 (m, 6H, 2NCH_2_CH_3_, *J* 7.2 Hz). ^13^C NMR (CDCl_3_, 100 MHz) δ: 167.0, 166.9 (COOCH_3_), 162.0 (C4), 151.9 (C6), 150.8 (C2), 145.9, 145.6, 129.4, 129.3, 128.6, 128.5, 128.4, (Ar1-Ar6, Ar1′-Ar6′), 101.6 (C5), 57.9, 57.8 (C10, C10′), 53.0, 52.8 (C11, C11′), 51.9, 51.8 (NCH_2_CH_3_), 47.5, 47.3 (C7, C7′), 44.9 (COOCH_3_), 26.76, 25.4, 24.5 (C9, C9′, C8, C8′), 19.5 (C6CH_3_), 11.7 (NCH_2_CH_3_). MALDI-MS (*m/z*): calcd for C_35_H_48_N_4_O_6_ [*M*]^+^ 620.4, found: 620.0. Anal. Calcd for C_33_H_44_N_4_O_6_: C, 67.72; H, 7.79; N, 9.03. Found: C, 67.76; H, 7.85; N, 8.95.

*1,3-Bis[5-(para-methoxycarbonylbenzylethylamino)pentyl]-6-methyluracil (**3b**)*: Yield 69%; oil; ^1^H NMR (CDCl_3_, 400 MHz) δ 7.98–7.95, 7.41–7.39 (both m, 4H each, 8H, Ar3H, Ar5H, Ar3′H, Ar5′H, Ar2H, Ar6H, Ar2′H, Ar6′H), 5.56 (s, 1H, C5H), 3.91 (br. s, 6H, 2COOCH_3_), 3.90–3.88 (m, 2H, N3C7′H_2_), 3.77–3.73 (m, 2H, N1C7H_2_), 3.58 (c, 4H, C12CH_2_, C12′CH_2_), 2.53–2.45, 2.43–2.39 (both m, 4H each, 8H, C11CH_2_, C11′CH_2_, 2NCH_2_CH_3_), 2.19 (s, 3H, C6CH_3_), 1.60–1.51, 1.50, 1.34–1.30 (all m, 4H each, 12H, C8CH_2_, C9CH_2_, C10CH_2_, C8′CH_2_, C9′CH_2_, C10′CH_2_), 1.04–0.99 (m, 6H, 2NCH_2_CH_3_). ^13^C NMR (CDCl_3_, 100 MHz) δ: 166.9, 166.8 (COOCH_3_), 162.0 (C4), 151.8 (C6), 150.8 (C2), 145.4, 129.3, 128.6, 128.5 (Ar1-Ar6, Ar1′-Ar6′), 101.4 (C5), 57.7, 57.6 (C11, C11′), 52.9 (C12, C12′), 51.8, 51.7 (NCH_2_CH_3_), 47.4, 47.2 (C7, C7′), 44.9 (COOCH_3_), 28.6, 27.2, 24.5, 24.3, 24.5 (C10, C10′, C9, C9′, C8, C8′), 19.5 (C6CH_3_), 11.5, 11.4 (NCH_2_CH_3_). MALDI-MS (*m/z*): calcd for C_37_H_52_N_4_O_6_ [*M*-H]^+^ 647.4, found: 647.6. Anal. Calcd for C_37_H_52_N_4_O_6_: C, 68.49; H, 8.08; N, 8.64. Found: C, 68.60; H, 8.00; N, 8.57.

*1,3-Bis[6-(para-methoxycarbonylbenzylethylamino)hexyl]-6-methyluracil (**3c**)*: Yield 70%; oil; ^1^H NMR (CDCl_3_, 400 MHz) δ 7.93, 7.37 (both m, 4H each, 8H, Ar3H, Ar5H, Ar3′H, Ar5′H, Ar2H, Ar6H, Ar2′H, Ar6′H), 5.52 (s, 1H, C5H), 3.87 (br. s, 8H, N3C7′H_2_, 2COOCH_3_), 3.72 (m, 2H, N1C7H_2_), 3.55 (br. s, 4H, C12CH_2_, C12′CH_2_), 2.46, 2.37 (both m, 4H each, 8H, C12CH_2_, C12′CH_2_, 2NCH_2_CH_3_), 2.17 (s, 3H, C6CH_3_), 1.57, 1.42 (both m, 4H each, C8CH_2_, C11CH_2_, C8′CH_2_, C11′CH_2_), 1.28 (m, 8H, C9CH_2_, C10CH_2_, C9′CH_2_, C10′CH_2_), 0.98 (m, 6H, 2NCH_2_CH_3_). ^13^C NMR (CDCl_3_, 100 MHz) δ: 167.0 (COOCH_3_), 162.1 (C4), 151.9 (C6), 150.8 (C2), 145.8, 129.3, 128.5 (Ar1-Ar6, Ar1′-Ar6′), 101.7 (C5), 58.0, 57.9 (C12, C12′), 53.3, 53.1 (C13, C13′), 51.9, 51.8 (NCH_2_CH_3_), 47.3, 47.2 (C7, C7′), 45.0 (COOCH_3_), 28.8, 27.5, 27.0, 26.9, 26.8, 26.7, 26.5 (C11, C11′, C10, C10′, C9, C9′, C8, C8′), 19.6 (C6CH_3_), 11.7, 11.6 (NCH_2_CH_3_). MALDI-MS (*m/z*): calcd for C_39_H_56_N_4_O_6_ [*M*]^+^ 676.4, found: 676.1. Anal. Calcd for C_39_H_56_N_4_O_6_: C, 69.20; H, 8.34; N, 8.28. Found: C, 69.31; H, 8.40; N, 8.22.

*1,3-Bis[4-(meta-methoxycarbonylbenzylethylamino)butyl]-quinazoline-2,4-dione (**4a**):* Yield 51%; oil; ^1^H NMR (CDCl_3_, 400 MHz) δ 8.20–8.18 (d, 1H, *J* 7.8 Hz, Ar1′’H), 7.96, 7.94 (both s, 1H each, 2H, Ar2H, Ar2′H), 7.89–7.86 (m, 2H, Ar5H, Ar5′H), 7.63–7.52 (m, 3H, Ar3′’H, Ar4H, Ar4′H), 7.36–7.31, 7.22–7.16 (both m, 2H each, 4H, Ar6H, Ar6′H, Ar2″H, Ar1″H), 4.08–4.03 (m, 4H, N3C7′H_2_, N1C7H_2_), 3.87, 3.86 (both s, 3H each, 6H, 2COOCH_3_), 3.59 (br. s, 4H, C11CH_2_, C11′CH_2_), 2.49 (m, 8H, C11CH_2_, C11′CH_2_, 2NCH_2_CH_3_), 2.18 (s, 3H, C6CH_3_), 1.76–1.65, 1.63–1.53 (both m, 4H each, 8H, C8CH_2_, C9CH_2_, C8′CH_2_, C9′CH_2_), 1.04–1.00 (m, 6H, 2NCH_2_CH_3_). ^13^C NMR (CDCl_3_, 100 MHz) δ: 167.1, 167.0 (COOCH_3_), 161.6 (C4), 150.6 (C2), 140.3 (C6), 139.7, 130.1, 130.0, 129.8, 129.8, 129.0, 128.2, 128.1, 127.9, 127.8 (Ar1-Ar6, Ar1′-Ar6′, Ar1″, Ar3″), 122.6 (Ar2″), 115.7 (C5), 113.6 (Ar4″), 57.7, 57.6 (C10, C10′), 52.8, 52.6 (C11, C11′), 52.0, 51.9 (NCH_2_CH_3_), 47.3, 47.2 (C7, C7′), 43.5 (COOCH_3_), 25.7, 25.0, 24.4, 24.3 (C9, C9′, C8, C8′), 11.6, 11.5 (NCH_2_CH_3_). MALDI-MS (*m/z*): calcd for C_38_H_48_N_4_O_6_ [*M*-H]^+^ 655.4, found: 655.3. Anal. Calcd for C_38_H_48_N_4_O_6_: C, 69.49; H, 8.53; N, 9.03. Found: C, 67.80; H, 8.42; N, 8.98.

*1,3-Bis[4-(meta-methoxycarbonylbenzylethylamino)pentyl]-quinazoline-2,4-dione (**4b**):* Yield 59%; oil; ^1^H NMR (CDCl_3_, 400 MHz) δ 8.21–8.19 (d, 1H, *J* 8.0 Hz, Ar1′’H), 7.97, 7.94 (both s, 1H each, 2H, Ar2H, Ar2′H), 7.89–7.86 (m, 2H, Ar5H, Ar5′H), 7.64–7.59 (m, 1H, Ar3′’H), 7.54–7.51 (both m, 2H each, H, Ar4H, Ar4′H, Ar6H, Ar6′H), 7.22–7.18 (m, 1H, Ar2′’H), 7.13–7.11 (m, 1H, *J* 8.4 Hz, Ar1′’H), 4.08–4.02 (m, 4H, N3C7′H_2_, N1C7H_2_), 3.89, 3.88 (both s, 3H each, 6H, 2COOCH_3_), 3.57, 3.56 (both s, 2H each, 4H, C12CH_2_, C12′CH_2_), 2.52–2.45, 2.44–2.39 (both m, 4H each, 8H, C11CH_2_, C11′CH_2_, 2NCH_2_CH_3_), 2.18 (s, 3H, C6CH_3_), 1.72–1.63, 1.54–1.44, 1.39–1.33 (all m, 4H each, 12H, C8CH_2_, C9CH_2_, C10CH_2_, C8′CH_2_, C9′CH_2_, C10′CH_2_), 1.02–0.98 (m, 6H, 2NCH_2_CH_3_). ^13^C NMR (CDCl_3_, 100 MHz) δ: 167.2, 167.1 (COOCH_3_), 161.6 (C4), 150.5 (C2), 140.6 (C6), 139.7, 130.1, 130.0, 129.7, 128.1, 128.0, 127.9 (Ar1-Ar6, Ar1′-Ar6′, Ar1′’, Ar3′’), 122.5 (Ar2′’), 115.7 (C5), 113.4 (Ar4′’), 57.7, 57.6 (C11, C11′), 53.0, 52.9 (C12, C12′), 51.9, 51.8 (NCH_2_CH_3_), 47.3, 47.1 (C7, C7′), 43.6 (COOCH_3_), 27.6, 27.1, 26.7, 26.6, 24.8, 24.5 (C10, C10′, C9, C9′, C8, C8′), 11.6 (NCH_2_CH_3_). MALDI-MS (*m/z*): calcd for C_40_H_52_N_4_O_6_ [*M*-H]^+^ 684.4, found: 683.3. Anal. Calcd for C_40_H_52_N_4_O_6_: C, 70.15; H, 7.65; N, 8.18. Found: C, 70.06; H, 7.72; N, 8.27.

#### 3.1.3. Conversion of 6-Methyl Derivatives with Methyl Benzoate Moieties in Salt and Acid Forms

*Preparation of salts **2e**, **3d***. *General procedure*: A mixture of methyl benzoate **2c**, **3b** (3.0 mmol) and sodium alkali (0.60 g, 15 mmol) was stirred in MeOH (40 mL) under reflux for 5 h. The aimed salts were isolated by column flash chromatography with MeOH as eluent.

*1,3-Bis[5-((meta-methylbenzoate)ethylamino)pentyl]-6-methyluracil disodium salt (**2e**)*: Yield 70%; m. *p*. 120–122 °C; ^1^H NMR (DMSO-d_6_, 600 MHz) δ 7.85, 7.83 (both s, 1H each, 2H, Ar2H, Ar2′H), 7.78–7.76 (m, 2H, Ar5H, Ar5′H), 7.26–7.24, 7.22–7.19 (both m, 2H each, 4H, Ar4H, Ar4′H, Ar6H, Ar6′H), 5.55 (s, 1H, C5H), 3.75–3.70 (m, 4H, N3C7′H_2_, N1C7H_2_), 3.50, 3.49 (both s, 2H each, 4H, C12CH_2_, C12′CH_2_), 2.44–2.39, 2.37–2.33 (both m, 4H each, 8H, C11CH_2_, C11′CH_2_, 2NCH_2_CH_3_), 2.21 (s, 3H, C6CH_3_), 1.52–1.46, 1.44–1.40 (both m, 4H each, 8H, C8CH_2_, C10CH_2_, C8′CH_2_, C10′CH_2_), 1.28–1.24, 1.22–1.19 (both m, 2H each, 4H, C9CH_2_, C9′CH_2_), 0.96–0.93 (m, 6H, 2NCH_2_CH_3_); (D_2_O, 600 MHz) δ 7.80–7.77 (m, 4H, Ar2H, Ar2′H, Ar5H, Ar5′H), 7.45–7.42 (m, 4H, Ar4H, Ar4′H, Ar6H, Ar6′H), 5.62 (s, 1H, C5H), 3.73–3.69 (m, 8H, N3C7′H_2_, N1C7H_2_, C12CH_2_, C12′CH_2_), 2.60, 2.47 (both m, 4H each, 8H, C11CH_2_, C11′CH_2_, 2NCH_2_CH_3_), 2.16 (s, 3H, C6CH_3_), 1.48 (m, 8H, C8CH_2_, C10CH_2_, C8′CH_2_, C10′CH_2_), 1.17 (m, 4H, C9CH_2_, C9′CH_2_), 1.04 (m, 6H, 2NCH_2_CH_3_). ^13^C NMR (D_2_O, 150 MHz) δ: 174.4 (COONa), 164.2 (C4), 155.2 (C6), 152.3 (C2), 136.9, 135.4, 135.1, 132.7, 130.7, 128.7, 128.6, 128.5 (Ar1-Ar6, Ar1′-Ar6′), 100.8 (C5), 56.7, 56.6 (C11, C11′), 51. 8 (C12, C12′), 47.1, 47.0 (NCH_2_CH_3_), 45.2 (C7, C7′), 27.7, 26.7, 24.3, 24.2, 24.1, 23.9 (C10, C10′, C9, C9′, C8, C8′), 19.1 (C6CH_3_), 9.8, 9.7 (NCH_2_CH_3_). MALDI-MS (*m/z*): calcd for C_35_H_46_N_4_Na_2_O_6_ [*M*+H]^+^ 665.3, [*M*+Na]^+^ 687.3, found: 665.8, 687.7. Anal. Calcd for C_35_H_46_N_4_Na_2_O_6_: C, 63.24; H, 6.98; N, 8.43; Na, 6.92. Found: C, 63.17; H, 7.06; N, 8.51; Na, 7.02.

*1,3-Bis[5-((para-methylbenzoate)ethylamino)pentyl]-6-methyluracil disodium salt (**3d**)*: Yield 72%; decomp. > 240 °C; ^1^H NMR (DMSO-d_6_, 600 MHz) δ 7.81–7.78, 7.18–7.15 (both m, 4H each, 8H, Ar3H, Ar5H, Ar3′H, Ar5′H, Ar2H, Ar6H, Ar2′H, Ar6′H), 5.58 (s, 1H, C5H), 3.77–3.69 (m, 4H, N3C7′H_2_, N1C7H_2_), 3.48, 3.47 (both s, 2H each, 4H, C12CH_2_, C12′CH_2_), 2.43–2.38, 2.37–2.29 (both m, 4H each, 8H, C11CH_2_, C11′CH_2_, 2NCH_2_CH_3_), 2.19 (s, 3H, C6CH_3_), 1.54–1.39 (m, 8H, C8CH_2_, C10CH_2_, C8′CH_2_, C10′CH_2_), 1.27–1.18 (m, 4H 4H, C9CH_2_, C9′CH_2_), 0.97–0.92 (m, 6H, 2NCH_2_CH_3_); (D_2_O, 600 MHz) δ 7.82–7.80, 7.28–7.26 (both m, 4H each, 8H, Ar3H, Ar5H, Ar3′H, Ar5′H, Ar2H, Ar6H, Ar2′H, Ar6′H), 5.61 (s, 1H, C5H), 3.77–3.75 (m, 4H, N3C7′H_2_, *J* 6.6 Hz), 3.70–3.68 (m, 4H, N1C7H_2_, *J* 6.0 Hz), 3.50 (s, 4H, C12CH_2_, C12′CH_2_), 2.42–2.40, 2.32–2.29 (both m, 4H each, 8H, C11CH_2_, C11′CH_2_, 2NCH_2_CH_3_), 2.15 (s, 3H, C6CH_3_), 1.49–1.40 (m, 8H, C8CH_2_, C10CH_2_, C8′CH_2_, C10′CH_2_), 1.15–1.13 (m, 4H 4H, C9CH_2_, C9′CH_2_), 0.96–0.94 (m, 6H, 2NCH_2_CH_3_). ^13^C NMR (D_2_O, 150 MHz) δ: 174.8 (COOCH_3_), 164.4 (C4), 154.9 (C6), 152.4 (C2), 140.4, 140.3, 135.7, 129.6, 129.2 (Ar1-Ar6, Ar1′-Ar6′), 101.8 (C5), 56.8, 56.7 (C11, C11′), 52.2, 52.1 (C12, C12′), 47.0, 46.9 (NCH_2_CH_3_), 45.2 (C7, C7′), 28.0, 26.9, 24.8, 24.3, 24.1 (C10, C10′, C9, C9′, C8, C8′), 19.2 (C6CH_3_), 10.2 (NCH_2_CH_3_). MALDI-MS (*m/z*): calcd for C_35_H_46_N_4_Na_2_O_6_ [*M*-H]^+^ 665.3, [*M*+Na]^+^ 687.3, found: 665.7, 687.7. Anal. Calcd for C_35_H_46_N_4_Na_2_O_6_: C, 63.24; H, 6.98; N, 8.43; Na, 6.92. Found: C, 63.15; H, 7.05; N, 8.48; Na, 6.98.

*Preparation of acid **2f***: Solution of salt **2e** (1.0 g, 1.5 mmol) in EtOH (20 mL) was transferred to a column with Dowex^®®^ G26 hydrogen form in EtOH. The aimed acid was obtained by passing through the column.

*1,3-Bis[5-(meta-carboxylbenzylethylamino)pentyl]-6-methyluracil (**2f**)*: Yield 99%; foam; ^1^H NMR (DMSO-d_6_, 400 MHz) δ 7.84, 7.82 (both s, 1H each, 2H, Ar2H, Ar2′H), 7.76–7.73 (m, 2H, Ar5H, Ar5′H), 7.25–7.18 (m, 4H, Ar4H, Ar4′H, Ar6H, Ar6′H), 5.57 (s, 1H, C5H), 3.76–3.69 (m, 4H, N3C7′H_2_, N1C7H_2_), 3.49, 3.48 (both s, 2H each, 4H, C12CH_2_, C12′CH_2_), 2.45–2.39, 2.37–2.31 (both m, 4H each, 8H, C11CH_2_, C11′CH_2_, 2NCH_2_CH_3_), 2.21 (s, 3H, C6CH_3_), 1.53–1.40 (m, 8H, C8CH_2_, C10CH_2_, C8′CH_2_, C10′CH_2_), 1.29–1.17 (m, 4H, C9CH_2_, C9′CH_2_), 0.97–0.92 (m, 6H, 2NCH_2_CH_3_); (D_2_O, 600 MHz) δ 7.77–7.75, 7.31–7.28 (both m, 4H each, 8H, Ar2H, Ar2′H, Ar5H, Ar5′H, Ar4H, Ar4′H, Ar6H, Ar6′H), 5.56 (s, 1H, C5H), 3.72–3.68, 3.64–3.60 (both m, 2H each, 4H, N3C7′H_2_, N1C7H_2_), 3.56, 3.55 (both s, 2H each, 4H, C12CH_2_, C12′CH_2_), 2.48–2.44, 2.36–2.31 (both m, 4H each, 8H, C11CH_2_, C11′CH_2_, 2NCH_2_CH_3_), 2.10 (s, 3H, C6CH_3_), 1.42–1.32 (m, 8H, C8CH_2_, C10CH_2_, C8′CH_2_, C10′CH_2_), 1.12–1.00 (m, 4H, C9CH_2_, C9′CH_2_), 0.98–0.94 (m, 6H, 2NCH_2_CH_3_). ^13^C NMR (DMSO-d_6_, 100 MHz) δ: 170.5 (COOH), 161.7 (C4), 153.0 (C6), 151.9 (C2), 139.7, 139.2, 129.9, 129.7, 129.6, 127.9, 127.3 (Ar1-Ar6, Ar1′-Ar6′), 100.5 (C5), 58.0, 57.9 (C11, C11′), 52. 9, 52.8 (C12, C12′), 47.2, 47.0 (NCH_2_CH_3_), 44.8 (C7, C7′), 28.4, 27.4, 26.6, 26.5, 24.6, 24.5 (C10, C10′, C9, C9′, C8, C8′), 19.5 (C6CH_3_), 12.1, 12.0 (NCH_2_CH_3_); (D_2_O, 150 MHz) δ: 175.5 (COOH), 164.7 (C4), 155.5 (C6), 152.3 (C2), 137.3, 136.8, 136.6, 133.0, 131.0, 129.0, 128.8 (Ar1-Ar6, Ar1′-Ar6′), 101.7 (C5), 57.3, 57.2 (C11, C11′), 52.4 (C12, C12′), 47.4, 47.3 (NCH_2_CH_3_), 45.6 (C7, C7′), 28.3, 27.3, 25.0, 24.9, 24.8, 24.5 (C10, C10′, C9, C9′, C8, C8′), 10.6, 10.5 (C6CH_3_), 10.5 (NCH_2_CH_3_). MALDI-MS (*m/z*): calcd for C_35_H_48_N_4_O_6_ [*M*+H]^+^ 621.4, [*M*+Na]^+^ 643.4, found: 621.3, 643.3. Anal. Calcd for C_35_H_48_N_4_O_6_: C, 67.72; H, 7.79; N, 9.03. Found: C, 67.66; H, 7.86; N, 8.96.

### 3.2. Molecular Modeling

The geometry of the ligand was quantum-mechanically (QM) optimized in the Gamess-US package [21] using the B3LYP DFT method and 6–31G* basis set. X-ray crystallographic structure of hAChE (PDB ID 4EY4) [22] was used as target for molecular docking. Molecular docking with a Lamarckian Genetic Algorithm (LGA) [23] was performed with Autodock 4.2.6 [24]. The grid box for docking included the whole active site and the gorge of AChE (22.5 × 22.5 × 22.5 Å grid box dimensions) with a grid spacing of 0.375 Å. The main LGA parameters were 256 runs, 25 × 10^6^ evaluations, 27 × 10^4^ generations, and a population size of 300. Figures were prepared with PyMOL (Schrödinger, LLC). 

### 3.3. Biological Studies

#### 3.3.1. In Vitro Cholinesterase Inhibition Assay

The inhibitory potency of compounds against hAChE and hBChE was studied using Ellman’s method [15]. Acetylthiocholine iodide (ATC), butyrylthiocholine iodide (BTC), recombinant hAChE, hBChE from blood plasma, and 5,5′-dithio-bis-(2-nitrobenzoic) acid (DTNB) were purchased from Sigma-Aldrich (Sigma-Aldrich, St. Louis, MO, USA). Stock solutions of compounds (0.01 M) were prepared using ethanol or water. In the case of compounds soluble in ethanol, the final concentration of ethanol in the cuvette was 0.1% vol. All assays were performed at 25 °C using a UV-1800 Shimadzu (Kyoto, Japan) spectrophotometer at 412 nm. The enzyme-catalyzed reaction of hydrolysis was carried out in 0.1 M phosphate buffer at pH 8.0 containing 0.25 units of hAChE or hBChE and 1 mM ATC or BTC as substrates. The tested compounds were pre-incubated with the enzyme for 5 min at 25 °C prior to the addition of the substrate and the commencement of hydrolysis kinetics recording. Experiments were performed in triplicate. The rate of substrate hydrolysis was calculated as measured by optical density change at 412 nm change over 2 min. The sample without substrate was used as a blank. The sample without inhibitor was used as a control (in a solution of 0.1% vol. ethanol in the case of compounds soluble in ethanol). Percentage of AChE/BChE inhibition was determined by comparison of rates of reaction of test samples relative to the control sample. The percentage of inhibition as a function of compound concentration was plotted using OriginPro 8.5 software (OriginLab Corporation, Northampton, MA, USA). IC_50_ (the concentration of the compound producing 50% enzyme activity inhibition) was calculated using the Hill Equation (1):(1)$$\frac{E}{E_{max}}=\frac{{\left[I\right]}^{n}}{IC_{50}^{n}+{\left[I\right]}^{n}},$$

The inhibition constants (Ki) for hAChE and hBChE were determined by Ellman’s method [15] at 412 nm and 25 °C in 0.1M sodium phosphate buffer (pH = 8.0) with various substrate (ATC, BTC) concentrations in the range 0.01–0.1 mM, using UV-1800 Shimadzu (Kyoto, Japan) spectrophotometer. The final enzyme concentration in the assays was in the picomolar range. Controls of active enzyme and substrate spontaneous hydrolysis in the absence of an inhibitor were performed for each substrate concentration. Absorbance change was recorded for 5 min. Inhibition constants were determined from Dixon plot and Cornish-Bowden transformation [25].

#### 3.3.2. Analysis of Compound **2c** Stability in the Presence of AChE

The compound **2c** was dissolved in ethanol and then in 2 mL of 0.1M sodium phosphate buffer (pH = 8.0) at the final concentration of 0.1 mM. Then, 2.0 units of hAChE were added. The final concentration of ethanol in the buffer was 0.1% vol. The compound **2c** was incubated with hAChE overnight at 25 °C. The same volume of buffer containing compound **2c** at the concentration of 0.1 mM but without AChE was used as a control. After the end of incubation, the buffer was filtered out of AChE using a centricon-30 ultrafiltration micro-concentrator from Amicon (Millipore Corporation, Billerica, MA, USA). Samples of the phosphate buffer were evaporated in vacuum, 1 mL of deuterated chloroform was added to the residue, and the mixture was filtered and transferred to a 5 mm ampoule to register NMR spectra.

#### 3.3.3. Acute Toxicity Evaluation and Brain AChE Inhibition Assay

All experiments involving animals were performed in accordance with the guidelines set out by the European Union Council Directive 2010/63/EU; the experimental protocols were approved by the Animal Care and Use Committee of the Federal Research Center “Kazan Scientific Center of the Russian Academy of Sciences”. Toxicological experiments were performed using intraperitoneal injection of the various compounds in CD1 mice weighing 20–25 g. Mice were maintained on a 12 h light/dark cycle (light from 7:00 a.m. to 7:00 p.m.) at 20−22 °C and 60−70% relative humidity. Five different doses (determined during preliminary tests) were used with six animals per dose. Animals were observed 14 days after injection, and symptoms of intoxication were recorded. LD_50_ dose (in mg/kg) causing lethal effects in 50% of animals was taken as a criterion of toxicity. LD_50_ was determined by the method of Weiss [26]. Stock solutions of compounds were prepared in ethanol or water. Compounds were administered intraperitoneally in sterile physiological saline. The concentration of ethanol in physiological saline was 5% vol.

For brain AChE inhibition assay, brains of mice were removed 30 min after i.p. injection of the compound **2c** at the LD_50_ dose (experimental group, n = 6 mice) or following i.p. vehicle injection (control group, n = 6 mice). The brains were subsequently frozen in liquid nitrogen. Whole brain homogenates were prepared in a Potter homogenizer with 0.05 M Tris-HCl, 1% Triton X-100, 1M NaCl, 2 mM ethylenediaminetetraacetic acid (pH 7.0; 4 °C; 1 part brain to 2 parts of buffer). The homogenate was centrifuged (14,000 × g rpm; t = 4 °C) for 10 min using an Eppendorf 5430R centrifuge with a FA-45-30-11 rotor (Eppendorf AG, Hamburg, Germany). AChE activity in brain homogenates was measured using the method described by Dingova et al. [27] with modification. Briefly, a 50 µL aliquot of supernatant was incubated for 30 min with 5 µL of 0.5 mM tetra-isopropyl pyrophosphoramide (iso-OMPA) as a specific BChE inhibitor. Next, the enzyme-catalyzed reaction was initiated by adding 10 µL of ATC (final concentration 1.5 mM) as a substrate. After 10, 20, or 30 min of incubation with the substrate at 25 °C, the reaction was stopped by adding neostigmine (final concentration 0.01 M). After diluting samples 25 times in a 50 mM phosphate buffer (pH 8.0), DTNB (0.1 mM) was added. The production of yellow 5-thio-2-nitro-benzoate anion resulting from the reduction of DTNB by thiocholine (the product of enzymatic hydrolysis of ATC) was measured spectrophotometrically according to the Ellman’s method [15]. The rate of thiocholine production over 20 min (from the 10th to the 30th min) was calculated. Brain samples of the control group were used as a control (100% of cholinesterase activity). The sample without substrate was used as a blank. All measurements for each brain sample were performed in triplicate.

#### 3.3.4. Animals and Treatments

##### Scopolamine Mouse Model

Scopolamine, representing a muscarinic acetylcholine receptor antagonist, produces a transient receptor blockade and cognitive deficits, which can be considered as a model of AD [28]. CD1 mice were randomly divided into 5 groups (n = 8–10): control group of mice treated with saline; mice injected with a water solution of scopolamine (2 mg/kg, i.p.) at 40 min before testing; mice treated with scopolamine (2 mg/kg, i.p.) and different dosages of compound **2c** (5 and 10 mg/kg p.o.) in water solution or donepezil (1 mg/kg, p.o.) at 40 and 20 min before testing, respectively. Drugs were administered over 18 days.

##### Transgenic Mouse Model

B6C3-Tg(APP695)85Dbo Tg(PSEN1)85Dbo double transgenic (APP/PS1) mice expressing a chimeric mouse/human amyloid precursor protein and a mutant of human presenilin-1. Both mutations are associated with early-onset AD [29,30]. APP/PS1 rodents develop Aβ deposits in the brain and memory impairment by 6–8 months of age [31]. The APP/PS1 mouse line was purchased from Jackson Laboratories (Bar Harbor, ME, USA) and bred at the Puschino animal facility (Moscow region, Russia) branch of the Shemyakin and Ovchinnikov Institute of Bioorganic Chemistry (Moscow, Russia). APP/PS1 mice at the age of 6 months were delivered to the Arbuzov Institute of Organic and Physical Chemistry, where the mice were housed under standard laboratory conditions. Mice were assigned to four groups (n = 8–10). The compound **2c** or donepezil were administrated as described above for scopolamine mouse model. APP/PS1 mice and their wild-type non-transgenic littermates that received saline were used as positive and negative controls.

##### Memory Performance Study

A reward alternation task was used to assess spatial memory performance in T-maze (OpenScience, Moscow, Russia) as previously described [32,33]. Briefly, prior to commencing T-maze testing, mice were placed on a food-deprivation schedule over 3 days. On the fourth day, mice were placed in the start arm of the T-maze and made it possible to move freely. Over the next 14 days of testing, mice were given 6 pairs of training trials. In the first trial of each pair (forced trial), one of the goal arm guillotine doors was closed and the mouse was constrained to selecting the opposite arm. The mouse was returned to the start box 15–20 s after consuming the reward (diluted sweetened condensed milk). In the second (free choice) trial, both goal arm doors were opened, the arm opposite the one selected in the forced trial was baited. The criterion for a mouse having learned the rewarded alternation task was 3 consecutive days of at least 5 correct choices out of the 6 free trials. T-maze was thoroughly cleaned with 70% EtOH between tests. The percentages of correct choices and mice reaching the criterion of learning were evaluated along with the dynamic of reaching the criterion of learning.

##### Aβ Plaques Quantification

After behavioral testing, all the mice were deeply anesthetized with isoflurane and exsanguinated by transcardial perfusion with cold phosphate-buffered saline (0.1 M, pH 7.4), followed by 4% paraformaldehyde in 0.1 M phosphate buffer (0.1 M, pH 7.4). After removing whole brains, the tissues were post-fixed for 24 h in 4% paraformaldehyde (Panreac, Spain) and stored at 4 °C in 30% sucrose in phosphate buffer (0.1 M, pH 7.4), containing 0.01% sodium azide. Cross-sections (20 μm thick) were obtained using a motorized cryostat Microm HM525 (Thermo Scientific, Waltham, MA, USA). To stain Aβ plaques, 1% Thioflavin S (Sigma-Aldrich, USA) in 50% ethanol was applied for 5 min. Next, sections were incubated in 70° EtOH for 5 min, rinsed to remove EtOH and mounted in Immu-Mount (Sigma-Aldrich, USA). The images were observed by a confocal laser scanning microscope Leica TSC SP5 MP (Leica Microsystems, Wetzlar, Germany). A total of eight sections for each brain and four fields for each section were analyzed in the entorhinal cortex area. The mean thioflavin S fluorescent spots per mm^2^ and the mean area of thioflavin S fluorescent spots were analysed by using Leica LAS AF Lite and ImageJ software.

##### Statistical Analysis

Statistical analyses were performed using OriginPro 8.5 software (OriginLab Corporation, Northampton, MA, USA) and SPSS Statistics 22.0 (IBM Corp, Armonk, NY, USA). The data were reported as the Mean ± SE; *p*-values < 0.05 were considered statistically significant.

## 4. Conclusions

A series of AChE inhibitors based on 1,3-bis[ω-(substituted benzylethylamino)alkyl]uracils was expanded by synthesis and biological evaluation in vitro and in vivo of 6-methyluracil and quinazoline-2,4-dione derivatives with *meta*- and *para*-benzoate moieties at benzene rings within benzyl fragments. In in vitro experiments, some of the synthesized benzoates demonstrated affinity against AChE in a nanomolar range of concentrations with selectivity indexes for AChE over BChE three and more orders of magnitude. The lead compound **2c** prevented scopolamine-induced spatial memory impairment and significantly reduced memory impairments in the APP/PS1 mice. Administration of **2c** was shown to significantly decrease the Aβ plaque burden in the APP/PS1 mice. Thus, compound **2c** due to having disease-modifying action is of interest as a potential therapeutic agent for the treatment of AD.

## Data Availability

Not applicable.

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
