# Peer review of "Novel Uracil-Based Inhibitors of Acetylcholinesterase with Potency for Treating Memory Impairment in an Animal Model of Alzheimer’s Disease"

_molecules, 2022, doi:10.3390/molecules27227855_

Round 1

Reviewer 1 Report

The paper "Novel Uracil-Based Inhibitors of Acetylcholinesterase with Potency to Treat Memory Impairment in Animal Model of Alzheimer’s Disease", represents a continuation of the authors’ previously published studies on uracil-based inhibitors of human cholinesterases. In the paper, the authors deal with the influence of different substituents on uracil-based inhibitors, introduced into the structure with the aim to enhance the solubility of the compounds in water, with regard to the inhibition activity of those compounds towards acetyl- and butyrylcholinestrase. The whole idea of the paper, obtained results and discussion are well conceptualized, but in some parts of the paper an additional clarification is needed.  

In the Introduction part, an explanation why compounds with benzoate moieties with substituents in meta and para positions at the benzene ring within benzyl fragments are synthesised, and not those with substituents in ortho position, is missing. The whole paper represents a continuation of previous research on compounds with a similar structure with substituents in the ortho position at the benzene ring. Water solubility problem refers to those compounds, not the compounds with substituents in meta or para position. 

In the Results and Discussion part, lines 144-149. The sentence The aimed compounds with m-methyl benzoate moieties 2a-d exhibited activity against AChE and BChE that clearly depends on the length of the polymethylene chains. If compound 2a with trimethylene chains binds to AChE with an IC50 five orders of magnitude higher than the nanomolar concentrations, and its activity towards BChE is even higher than that towards AChE, then the activity and selectivity towards AChE sharply increase with increasing chains length should be modified as this statement is only partially true; it depends on the length of up to 5 methyl units in the linker. Additional elongation of the linker for one methyl units decreased the inhibition activity compared to compound 2c. This statement should be general to correspond to further discussions of those results by the authors.

Have the values for donepezil inhibition activity and LD50 been determined in this study? If yes, please discuss them with those previously published. If not, please provide a corresponding reference.

Figure 3. would be more informative and easier to illustrate the position of the 2c if the water accessible surface of active site gorge were used.

Section 3.3.2. Specify how the ability of a compound to penetrate the BBB is actually calculated. The description of the measurement procedure is quite confusing.

Next, the enzyme-catalyzed hydrolysis reaction was initiated by adding 10 μL of acetylthiocholine (0.01 M) as a substrate (Specify the final concentration of the ATCh). After 10, 20, or 30 min of incubation with the substrate at 25°C, the reaction was stopped by adding neostigmine (final concentration 0.01 M). After diluting samples 25 times in a 50 mM phosphate buffer (pH 8.0), DTNB (0.1 mM) was added. The production of yellow 5-thio-2-nitro-benzoate anion resulting from the reduction of DTNB by thiocholine (the product of enzymatic hydrolysis of acetylthiocholine) was measured spectrophotometrically according to the Ellman method [15]. Were end points of reaction measured? The rate of thiocholine production over 20 min (from the 10th to the 30th min) was calculated. Confusing sentence!

Some specific comments:

In the introduction part, fourth paragraph (lines 50-66), the authors should state a corresponding reference after every statement (Lines 52 and 58), not only at the end of the paragraph. Please check reference no. 13. I think that Semenov et all would be more proper, 6-Methyluracil Derivatives as Bifunctional Acetylcholinesterase Inhibitors for the Treatment of Alzheimer’s Disease. ChemMedChem 2015,10,1863–1874, since the IC50 values to which the authors refer in the manuscript are contained therein. Corresponding references should be indicated also at the and of the sentences in lines 68, 80 and 95.

Line 159 Please put mark 3 in brackets or in italic, as they are confusing

Line 161. M&M state that measurements were performed in triplicates; please put mark 1 on the corresponding place in the table

Line 164. The decrease is not actually dramatic, it is only 2.5 times lower compared to that of compound 2b.

In my personal opinion, a major revision of the manuscript would significantly improve the quality of this manuscript and make it acceptable for publication.

Author Response

Author's Notes to Reviewer are attached.

Reviewer 2 Report

The paper by V.E. Semenov et al. describes a classic medicinal chemistry work aimed at the characterization of a newly synthesyzed series of human acetylcholinesterase (AChE) and butyrrylchoinesterase inhibitors with potential as anti Alzheimer’s chemotherapics.

The molecules belong to a class of 6-methyluracil derivatives on which the authors have previously reported extensively.

With respect to the approved Alzheimer’s drug donepezil, the lead compound of the new series herein under study showed improved inhibition potency against AChE, comparable ability to restore memory in mice models of Alzheimer’s and an improved ability to reduce the formation of Aϐ plaque in the brain of APP/PS1 mice.

The work, that include design, synthesis, in vitro, in silico, in ex-vivo ad in vivo characaterization of the inhibitors, overall is scientifically sound and provide an appreciable contribution to the field of cholinesterares inhibition. There are however few aspects that need to be addressed before publication:

1. The quality of English needs substantial improving through out the paper. It might be advisable that the paper is revised by a native English speaker.

2. The authors claim that the new inhibitors were designed to improve the water solubility of the parent compound(s). But it is not self-evident why the replacement of the nitrobenzyl group (compound 1a) with a methyl-benzoate group should increase appreciably the molecule solubility in water. I suggest that if data on solubility of new and parent compounds are available, they are presented and discussed. Or else, that the section on the rationale for the design of the new compounds is modified.

2. Although comparison between IC50s is not always straightforward, it seems that 2c is an AChE inhibitor more than 150 times stronger than donepezil (Table 1). Yet, the efficacy of compound 2c to reverse the scopolamine induced amnesia (etc.) is somewhat weaker than that of donepezil (in molar terms, it is ~2-5 fold weaker, Figures 4,5).

In principle, this might depend on the pharmacodynamic of the two molecules, i.e., different binding to blood proteins, different rates of excretion/metabolization or different BBB crossing efficiency.

I understand that the pharmacodynamic characterization of the inhibitors is beyond the scope of the present work, but still, this aspect is too important to be neglected. In my opinion it deserves to be at least highlighted and commented in the conclusions section.

Moreover, even though the steady-state analysis of inhibitors is not an indispensable prerequisite for publication, the proper determination of the Ki against AChE, at least for compound 2c, would allow a more meaningful comparison with donepezil, hence improving the overall quality of the manuscript.

3. Oddly enough, authors have chosen to synthesize esters as inhibitors of an esterase.

The automated docking simulation of 2c into the human AChE active site suggests that, similarly to what was found in the crystallographic structure of the AChE-1a complex, the benzyl ring is positioned within the CAS.

On the basis of the docking pose showed in Figure 3, the authors state that:

the location of the ether oxygen atom makes proton transfer from His447 unlikely, along with hydrolysis of the methoxycarbonyl group per se” (lines: 214-216).

The catalytic mechanism, however, do not involve a direct His447 -substrate(s) interaction. Nor it is evident from Figure 3 that the binding of 2c has the potential to disrupt the catalytic triad. An event that, anyway, the automated docking is unable to predict, unless the side chain of His447 is specifically set to be treated as flexible by AutoDock.

So, the exact meaning of the quoted sentence is not clear.

Moreover, the docking of ligands with so many flexible bonds as 2c is always tricky, because the “right solution” has to be searched within a huge chemical space. This implies that the docking simulations have to be taken with much caution and, most importantly, that they can hardly be considered as an acceptable proof for the inability of AChE to hydrolyze these inhibitors.

Indeed, I find hard to believe that a methylester group so nicely positioned in the active site of an AChE can survive a long time before being hydrolyzed. Should this happen, even if only at a low rate, the inhibitors should actually be regarded as a substrates as well.

The catalytic processing of 2c would generate a product similar to compound 2e. Which is a ~50-fold weaker AChE inhibitor than 2c (Table 1) and, critically, only 3-4 times less potent than donepezil.

This, at least in principle, might explain the discrepancy between the IC50 and the in vivo efficacy of 2c compared to those of donepezil (see previous point).

To clear-up this doubt, authors should include data on the stability of 2c in the presence of AChE (in condition similar to that used for the in vitro assays). Determination of blood plasma stability of 2c would also help to put in context the in vivo data and to improve the paper.

Minor revisions:

Line 35: “catalase” should read “catalyze”.

Lines 43-45: the entire sentence should be rephrased.

Line 61: “X-ray data” should read “crystallographic data”. The PDB code and the associated reference should be provided.

Lines 146-149: the entire sentence should be rephrased.

Lines 219-221: I am afraid there is not any obvious relation between LD50 of a compound and its ability to cross the BBB.

Lines 224-225: It is not clear what is the 2c dose that caused a 52% reduction of the brain AChE activity.

Author Response

(The authors gave the same response as above.)

Reviewer 3 Report

Basic pharmacokinetic test data should be supported before the in vivo test, such as metabolic stability of liver microsome, plasma stability, plasma protein binding rate and BBB permeability. The pharmacodynamic results in vivo should match the pharmacokinetic data in vivo. It is suggested that the author supplement the pharmacokinetic data such as AUC and drug concentration curve in brain.

Author Response

(The authors gave the same response as above.)

Round 2

Reviewer 1 Report

The Authors responded to all queries and  improved overall presentation of results.

Reviewer 2 Report

The authors have satisfactory replied to all objections. The additional experimental data they have provided, together with the improvement of the English,  improved significantly the quality of the paper, that can now be published.

Reviewer 3 Report

It is hoped that the author will pay attention to the pharmacokinetic data during the in vivo test, such as brain blood ratio.